# Out-of-pocket expenditure on childhood infections and its financial burden on Indian households: Evidence from nationally representative household survey (2017–18)

Habib Hasan Farooqui[1], Anup Karan[2], Manu Raj Mathur[3,4]*, Suhaib Hussain[2], Sakthivel Selvaraj[4]

**1** College of Medicine, QU Health, Qatar University, Doha, Qatar, **2** Indian Institute of Public Health-Delhi, Gurugram, India, **3** Faculty of Medicine and Dentistry, Queen Mary University of London, London, United Kingdom, **4** Public Health Foundation of India, Gurugram, India

* m.r.mathur@qmul.ac.uk

**Data Availability Statement:** The datasets were derived from sources in the public domain: NSSO: Social Consumption and Health 75th round and

## Abstract

The key objective of this research was to estimate out of pocket expenditure (OOPE) incurred by the Indian households for the treatment of childhood infections. We estimated OOPE estimates on outpatient care and hospitalization by disease conditions and type of health facilities. In addition, we also estimated OOPE as a share of households' total consumption expenditure (TCE) by MPCE quintile groups to assess the quantum of the financial burden on the households. We analyzed the Social Consumption: Health (SCH) data from National Sample Survey Organization (NSSO) 75th round (2017–18). Outcome indicators were prevalence of selected infectious diseases in children aged less than 5 years, per episode of OOPE on outpatient care in the preceding 15 days, hospitalization in the preceding year and OOPE as a share of households' total consumption expenditure. Our analysis suggests that the most common childhood infection was 'fever with rash' followed by 'acute upper respiratory infection' and 'acute meningitis'. However, the highest OOPE for outpatient care and hospitalization was reported for 'viral hepatitis' and 'tuberculosis' episodes. Among the households reporting childhood infections, OOPE was 4.8% and 6.7% of households' total consumption expenditure (TCE) for outpatient care and hospitalization, respectively. Furthermore, OOPE as a share of TCE was disproportionately higher for the poorest MPCE quintiles (outpatient, 7.9%; hospitalization, 8.2%) in comparison to the richest MPCE quintiles (outpatient, 4.8%; hospitalization, 6.7%). This treatment and care-related OOPE has equity implications for Indian households as the poorest households bear a disproportionately higher burden of OOPE as a share of TCE. Ensuring financial risk protection and universal access to care for childhood illnesses is critical to addressing inequity in care.

can be downloaded upon registration at http://mospi.nic.in/unit-level-data-report-nss-75th-round-july-2017-june-2018-schedule-250social-consumption-health. All data generated or analysed during this study are included in this published article (and its supplementary information files).

**Funding:** The analysis and publication of this manuscript was supported by a Research England Policy Impact Funding. The Research England Policy Impact Funding was awarded to Queen Mary University of London and then subsequently allocated to a group of researchers on a competitive basis to publish policy relevant research. The specific grant allocation number was: PSF0002R.

**Competing interests:** The authors have declared that no competing interests exist.

## Introduction

Childhood infections remain one of the most common causes of childhood morbidity and mortality in India. Recent estimates suggest that the burden of infectious diseases has reduced in India in the past decade, but for five of the ten individuals, the leading causes of disease burden still belonged to infectious and associated diseases in the year 2016. Furthermore, the proportion of total disease burden caused by infectious and associated diseases is the highest among children [1]. Some of the reasons for high infection rates in India include not only demographic and epidemiological factors like high population densities, and high prevalence of risk factors such as malnutrition [2] but also economic determinants such as inadequately financed public health systems resulting into limited access to healthcare [3].

It may be noted that over the past two decades several public health measures have been implemented to reduce childhood morbidity and mortality in India. Some of them include the introduction of new vaccines in the universal immunization program [4], health system strengthening through National Health Mission [5, 6], and government-sponsored insurance schemes to improve access to hospital care [7] to provide financial risk protection to poor and less advantaged households. The latest evidence suggests that among all major causes of death in children in India, the decrease in the death rate was the highest for the infectious diseases [8] between 2000 and 2017.

However, childhood infections not only result in poor health outcome but also causes a severe financial burden on households as well as society. Furthermore, because of the absence of strong financial risk protection and social security measures in India, the majority of treatment and care-related expenditures are incurred as out of pocket expenditures (OOPE) by the households. These OOPEs usually turn catastrophic and impoverishing for households in India, especially in event of hospitalization [9]. One of the key reasons for the severe financial impact of infectious disease-related OOPE on households is uncertainty associated with the event, as the disease impacts or shock happens in a small time window, leaving a limited opportunity for the households to smoothen expenditure or initiate mitigation measures, especially in situations requiring hospitalization [10]. Literature suggests that households resort to measures such as dis-savings, borrowing, using loans or mortgages, and selling assets and livestock to meet OOPE [11]. It has also been argued that not only do poor households utilize less healthcare but also get disproportionately affected by the OOPE [12].

Although there is enough evidence on epidemiological measures of childhood infections such as morbidity pattern and mortality, the evidence on the treatment cost of childhood infectious diseases in terms of OOPE, its financial impact on households and its equity impact is lacking in India. Though previous community and hospital-based studies from India reported treatment costs for childhood infections such as acute respiratory infection [13] and diarrhoea [14, 15]. We recognized that most of the costing literature is focused on the cost of treatment of vaccine-preventable diseases such as pneumonia, diarrhoea, meningitis and others since treatment cost is one of the core parameters in the cost-effectiveness analysis of vaccines and other interventions. We also noted that previously published estimates on treatment cost of childhood infection are not truly representative of full financial cost and are not reflective of a true financial burden on households as they are context-specific and are generally focused on specific disease conditions. Furthermore, we identified that there were no published estimates on OOPE incurred by the households on several childhood infections such as fever with rash, viral hepatitis, diphtheria, and meningitis from India. In this context, the objective of this paper is to report per episode childhood infection-related OOPE on outpatient care and hospitalization in India. In addition, we also report the financial burden of the OOPE on households by estimating OOPE as a share of the total consumption expenditure using the latest available nationally representative household survey data (2017–18).

## Methods

### Data source and sample

We analyzed the Social Consumption: Health (SCH) data from National Sample Survey Organization (NSSO) 75th round (Schedule 25.0; July 2017 to June 2018) [16]. Nationally, ~1,13,823 households (64,552 in rural areas and 49,271 from urban areas) were included in the survey through a multistage stratified sampling process. The information is collected from selected households using a questionnaire schedule (25.0). The SCH (2017–18) survey was used for disease level classifications of OOPE in outpatient and hospitalisation care. In addition to a range of socio-economic identifiers, the SCH collected detailed information on the type of morbidities, health care utilization and expenditure pattern of the households associated with the self-reported morbidities and healthcare utilization separately in outpatient care and hospitalisation.

The NSSO schedule recorded the response of individuals/households to specific questions eliciting information on healthcare utilization and the reason for the same. In addition, the survey also asked the question, "What was the nature of ailment" classified by 60 different health conditions both for hospitalisation (365 days reference period) and outpatient (15 days reference) treatment. From these health conditions, infectious diseases for each individual can be identified. We matched the disease condition in the surveys to broad ICD disease classification. The selection of the childhood infectious diseases, in the present study, is driven by the prevalence of the disease, its potential economic burden and the availability of data on expenditure on treatment in the National Sample Survey Office (NSSO) data [16]. The working definition for the selected infectious diseases analysed in the research is provided in the S1 Table in S1 File.

SCH survey also separately records expenses incurred for hospitalisation and outpatient care with respective reference periods for each episode of treatment. The expenditure incurred by households was recorded for different heads of expenditure such as consultation charges, bed charges, purchase of medicines and diagnostic services. The survey also recorded information related to the money spent on transportation for accessing healthcare and other non-medical costs. Full details on different types of expenditures incurred by the household for the treatment of childhood infections are provided in the S2 Table in S1 File.

### Outcome measures

We reported (1) Prevalence of selected infectious diseases in children aged less than 5 years, (2) OOPE on outpatient care per episode in the preceding 15 days, (3) OOPE per hospitalization in the preceding year (4) OOPE as a share of households' total consumption expenditure and (5) OOPE as a share of households' total consumption expenditure by quintile groups, 5 equal groups of monthly per capita consumption expenditure (MPCE).

All outcomes were estimated and reported for children aged less than 5 years.

### Statistical analysis

We estimated the prevalence, healthcare utilisation, and mean OOPE per episode of outpatient visit and hospitalization in children aged less than 5 years for the selected infectious diseases. OOPE on outpatient care was estimated as a share of monthly household consumption expenditure and OOPE on hospitalization was estimated as a share of annual household consumption expenditure for each household reporting childhood infection.

We used household consumption expenditure as a denominator to OOPE, we standardized estimates of OOPE (across inpatient 365 days and outpatient 15 days) to 30 days. Since the

recall periods of inpatient outpatient expenditures are different (365 days and 15 days respectively), averaging for 30 days was essential to estimate the share of inpatient and outpatient expenses to total household consumption expenditure which is available only with 30 days reference period.

Data were analyzed using Stata software V.15.0 (StataCorp LP, College Station, Texas, USA) and p values of less than 0.05 were considered statistically significant. All analyses were carried out using sampling weight.

### Ethical approval

The study uses anonymized secondary data which is publicly available from the NSSO and hence doesn't involve any ethical issue.

## Results

### Prevalence and healthcare utilization

In terms of the prevalence, the most common ailments for which outpatient consultation was sought were fever with rash (44.0 per 1,000 children) followed by acute respiratory infections (16.8 per 1,000 children), and acute meningitis (6.2 per 1,000 children) (Table 1) whereas the

**Table 1. Prevalence and healthcare consultation for childhood infections (age 0–5 years) in outpatient and hospital settings.**

| Ailment | Outpatient care with 15-days recall | | | | Hospitalisation with 365-days recall | | |
|---|---|---|---|---|---|---|---|
| | Prevalence per 1,000 children (95% CI) | % Sought formal care (95% CI) | % treatment in public sector | % treatment in private sector | Hospitalisation per 1,000 children (95% CI) | % treatment in public sector | % treatment in private sector |
| Acute meningitis | 6.2 | 85.8 | 25.7 | 74.3 | 1.76 (1.45–2.06) | 41.1 | 58.9 |
| | (5.6–6.8) | (85.4–90.2) | | | | | |
| Fever due to diphtheria, whooping cough and tetanus | 6.1 | 78.3 | 33.2 | 66.8 | 0.68 (0.49–0.87) | 42.0 | 58.0 |
| | (5.5–6.7) | (73.9–82.6) | | | | | |
| Fever with rash * | 44.0 | 79.7 | 23.9 | 76.1 | 7.89 (7.23–8.54) | 32.6 | 67.4 |
| | (42.4–45.5) | (78.1–81.4) | | | | | |
| Tuberculosis | 0.1 | 100.0 | 11.7 | 88.3 | 0.06 (0.01–0.12) | 69.6 | 30.4 |
| | (0.02–0.16) | (100.0–100.0) | | | | | |
| Viral Hepatitis | 0.2 | 92.4 | 25.5 | 74.5 | 1.75 (1.44–2.05) | 41.3 | 58.7 |
| | (0.09–0.3) | (81.8–100.0) | | | | | |
| Acute diarrhea** | 3.5 | 80.7 | 30.8 | 69.2 | 2.32 (1.97–2.68) | 42.6 | 57.4 |
| | (3.1–3.9) | (75.6–85.8) | | | | | |
| Acute upper respiratory infections*** | 16.8 | 72.1 | 22.8 | 77.2 | 1.53 (1.24–1.81) | 38.1 | 61.9 |
| | (15.9–17.8) | (69.4–74.8) | | | | | |
| Fever with other infections# | 0.6 | 96.6 | 37.4 | 62.6 | 0.81 (0.59–1.01) | 33.7 | 66.4 |
| | (0.4–0.7) | (91.9–100.0) | | | | | |
| All Infectious diseases | 77.5 | 78.68 | 24.95 | 75.05 | 16.79 (15.85–17.74) | 36.86 | 63.14 |
| | (75.6–79.6) | (77.5–79.9) | | | | | |

Notes

* includes fevers of unknown origin, all specific fevers that do not have a confirmed diagnosis

** includes dysentery/ increased frequency of stools with or without blood and mucus in stools

*** includes cold, runny nose, sore throat with the cough, allergic colds included)

# includes malaria and worms infestations.

The numbers in parenthesis depict the confidence intervals

Source: Authors estimates using SCH 2017–18

hospitalization rates were the highest for fever with rash (7.9 per 1,000 children) followed by acute diarrhoea (2.3 per 1,000 children), viral hepatitis (1.76 per 1,000 children) and acute meningitis (1.75 per 1,000 children)

We also identified that the proportion of households who sought care at the public or private health care facility varied by disease condition and type of care (outpatient versus hospitalization). For example, except for tuberculosis, households preferred private sector health facilities over the public sector both for hospitalizations (range 57%-67%) as well as outpatient consultations (range 66%-88%).

Key socio-economic characteristics of all children (age<5 years), childhood infections and hospitalisations are presented in the S3 Table in S1 File. Socio-economic indicators suggest that poorer households have a proportionately larger concentration of children and they also reported a higher proportion of children ailing with infectious diseases as compared with those among richer households. Although overall reporting of childhood infections was higher among poor households, compared to richer households, utilization of hospital services was more concentrated in the richer households. Furthermore, children below the age of 2 years reported more ailments and higher healthcare utilization as compared with their share of the total children population. Also, households reported a higher share of childhood infections and higher utilization of healthcare for boys as compared to girls (S3 Table in S1 File). Further details about urban-rural differences in terms of prevalence, utilization and OOPE due to childhood infectious diseases are presented in the S5-S7 Tables in S1 File.

## Out-of-pocket expenditure on outpatient care and hospitalization

Per episode medical and non-medical expenditures incurred by the households on outpatient care in public versus private sector health facilities are reported in Table 2. Our estimates on outpatient treatment costs suggest that viral hepatitis accounted for the highest OOPE per episode (medical cost -INR 1478, non-medical cost -INR 362) followed by tuberculosis (medical cost -INR 1031, non-medical cost -INR 178). In general, across all disease conditions treatment costs incurred by households were higher in the private sector in comparison to the public sector. Private sector expenditure in outpatient care was driven by medicines where in

**Table 2. Per episode OOPE (INR) on outpatient care for childhood infections in India, 2018.**

| Ailment | Average OOPE per episode (INR) | | | | |
| --- | --- | --- | --- | --- | --- |
| | Medical (SE) | Non-medical (SE) | Total (SE) | Public (SE) | Private (SE) |
| Acute meningitis | 577 (43.1) | 100 (6.05) | 677 (45.5) | 184 (41.5) | 909 (60.9) |
| Fever due to diphtheria, whooping cough and tetanus | 504 (46.3) | 80 (6.2) | 584 (48.2) | 402 (41.0) | 718(64.5) |
| Fever with rash * | 437 (16.7) | 77 (3.6) | 514 (18.7) | 216 (14.7) | 661 (26.5) |
| Tuberculosis | 1031 (191.8) | 178 (47.0) | 1,209 (232.9) | 331 (128.8) | 1,324 (293.5) |
| Viral Hepatitis | 1478 (262.1) | 362 (117.8) | 1,840 (337.2) | 467 (299.6) | 2,323 (416.3) |
| Acute diarrhea** | 604 (58.3) | 73 (8.5) | 677 (64.2) | 539 (132.4) | 744 (88.05) |
| Acute upper respiratory infections*** | 343 (22.0) | 46 (3.1) | 389 (23.6) | 217 (15.3) | 495 (35.8) |
| Fever with other infections# | 714 (110.8) | 129 (19.4) | 843 (120.4) | 433 (113.0) | 1,091 (162.4) |

Notes

* includes fevers of unknown origin, all specific fevers that do not have a confirmed diagnosis

** includes dysentery/ increased frequency of stools with or without blood and mucus in stools

*** includes cold, runny nose, sore throat with cough, allergic colds included)

# includes malaria and worms infestations.

The numbers in parenthesis depict the standard errors

Source: Authors estimates using SCH 2017–18

**Table 3. Per episode OOPE (INR) on hospitalization for childhood infections, 2018.**

| Ailment | Average per episode OOPE (INR) | | | | |
|---|---|---|---|---|---|
| | Medical (SE) | Non-medical (SE) | Total (SE) | Public (SE) | Private (SE) |
| Acute meningitis | 5,367 (821.8) | 914 (95.7) | 6,281 (869.3) | 1,938 (302.8) | 9,313 (1434.1) |
| Fever due to diphtheria, whooping cough and tetanus | 7,590 (1423.1) | 1,332 (117.5) | 8,922 (1509.2) | 2,991 (309.1) | 13,212 (2741.5) |
| Fever with rash/ eruptive lesions* | 9,184 (712.9) | 1,200 (44.5) | 10,384 (730.5) | 5,518 (1754.1) | 12,740 (537.4) |
| Tuberculosis | 11,246 (6224.8) | 2,402 (640.8) | 13,648 (6796.5) | 3,490$ (943.1) | 9,364 (2555.4) |
| Viral Hepatitis | 12,546 (1413.2) | 1,378 (123.3) | 13,924 (1476.4) | 3,118 (430.9) | 21,527 (2462.9) |
| Acute diarrhea** | 5,174 (390.1) | 911 (41.9) | 6,085 (412.1) | 1,981 (196.4) | 9,136 (719.7) |
| Acute upper respiratory infections*** | 6,968 (781.3) | 1,362 (106.4) | 8,330 (820.5) | 3,216 (326.6) | 11,476 (1306.7) |
| Fever with other infections# | 6,292 (867.2) | 1,222 (90.07) | 7,514 (918.5) | 2,873 (720.8) | 9,867 (1441.2) |

Notes

* includes fevers of unknown origin, all specific fevers that do not have a confirmed diagnosis

** includes dysentery/ increased frequency of stools with or without blood and mucus in stools

*** includes cold, runny nose, sore throat with the cough, allergic colds included)

# includes malaria and worms infections

$ excluding one outlier case. The numbers in parenthesis depict the standard errors

Source: Authors estimates using SCH 2017–18

hospitalization it was driven by medicines and bed charges. However, in public sector the expenditure was driven by medicines for both outpatient care and hospitalization and non-medial expenditure such as food and lodging for patient and attendant in case of hospitalization. Further details about component-wise breakdown (such as doctor fee, medicine, diagnostics, travel, food, etc.) of the direct medical and non-medical expenditure, separately for public and private hospitals is provided in S8 and S9 Tables in S1 File.

Furthermore, our analysis suggests that viral hepatitis-related hospitalization (medical cost-INR 12,546, non-medical cost-INR 1,378) remained one of the highest expenditure items for households among all disease conditions, followed by tuberculosis (medical cost- INR 11,246, non-medical cost-INR 2,402) (Table 3)). Similar to outpatient treatment costs, the hospitalization costs were significantly higher in private sector health facilities (range INR 9,136-INR 21,527) as compared to the public sector (range INR 1938-INR 5518), for all disease conditions. Also, the proportion of non-medical costs ranged between 10% to 18% of the total treatment across disease conditions; lowest for viral hepatitis (10%) and highest for Tuberculosis and ARI (18%).

## Out-of-pocket expenditure by disease conditions and MPCE categories

Overall OOPE for childhood infections for outpatient care and hospitalisation constituted 4.8% and 6.7% of total household consumption expenditure respectively, among the households reporting such conditions (Table 4). The average share of household expenditure incurred on hospitalization was the highest for tuberculosis (12.5%) followed by viral hepatitis (10.9%) while the average share of household expenditure for outpatient care was highest for viral hepatitis (18.3%) followed by tuberculosis (10.3%). These OOPE often constitute a significantly high proportion of total consumption expenditure for many households. Figs 1 and 2 clearly demonstrate that childhood infections related to OOPE exert a severe financial burden on households as reflected in OOPE overshoot (defined as more than 10%, 20% or 40% of total household consumption expenditure and represented by red lines in Figs 1 and 2) in comparison to the mean overall household expenditure (a proxy for the household income). Among all

**Table 4. Average share of household expenditure incurred on childhood infections (age 0–5 years) in outpatient and hospitalization.**

| Ailment | OOPE share (%) on outpatient with 15-days recall (95%CI) | OOPE share (%) on hospitalisation with 365-days recall (95%CI) |
|---|---|---|
| Acute meningitis | 7.0 (5.6–8.3) | 4.7 (3.5–6.1) |
| Fever due to diphtheria, whooping cough and tetanus | 5.1 (4.1–6.0) | 6.5 (5.1–8.0) |
| Fever with rash* | 4.9 (4.4–5.3) | 7.2 (6.2–8.1) |
| Tuberculosis$ | 10.3 (7.4–13.1) | 12.5 (-0.4–25.5) |
| Viral Hepatitis | 18.3 (9.8–26.9) | 10.9 (8.2–13.8) |
| Acute diarrhea** | 7.1 (5.7–8.6) | 4.5 (4.0–5.3) |
| Acute upper respiratory infections*** | 3.4 (2.8–3.9) | 5.7 (4.5–7.0) |
| Fever with other infections# | 9.9 (6.5–13.1) | 6.5 (4.7–8.2) |
| Overall | 4.8 (4.5–5.1) | 6.7 (6.2–7.4) |

Notes

\* includes fevers of unknown origin, all specific fevers that do not have a confirmed diagnosis

\*\* includes dysentery/ increased frequency of stools with or without blood and mucus in stools

\*\*\* includes cold, runny nose, sore throat with the cough, allergic colds included)

# includes malaria and worms infestations

$ excluding one outlier case. The numbers in parenthesis depict the confidence intervals

Source: Authors estimates using SCH 2017–18

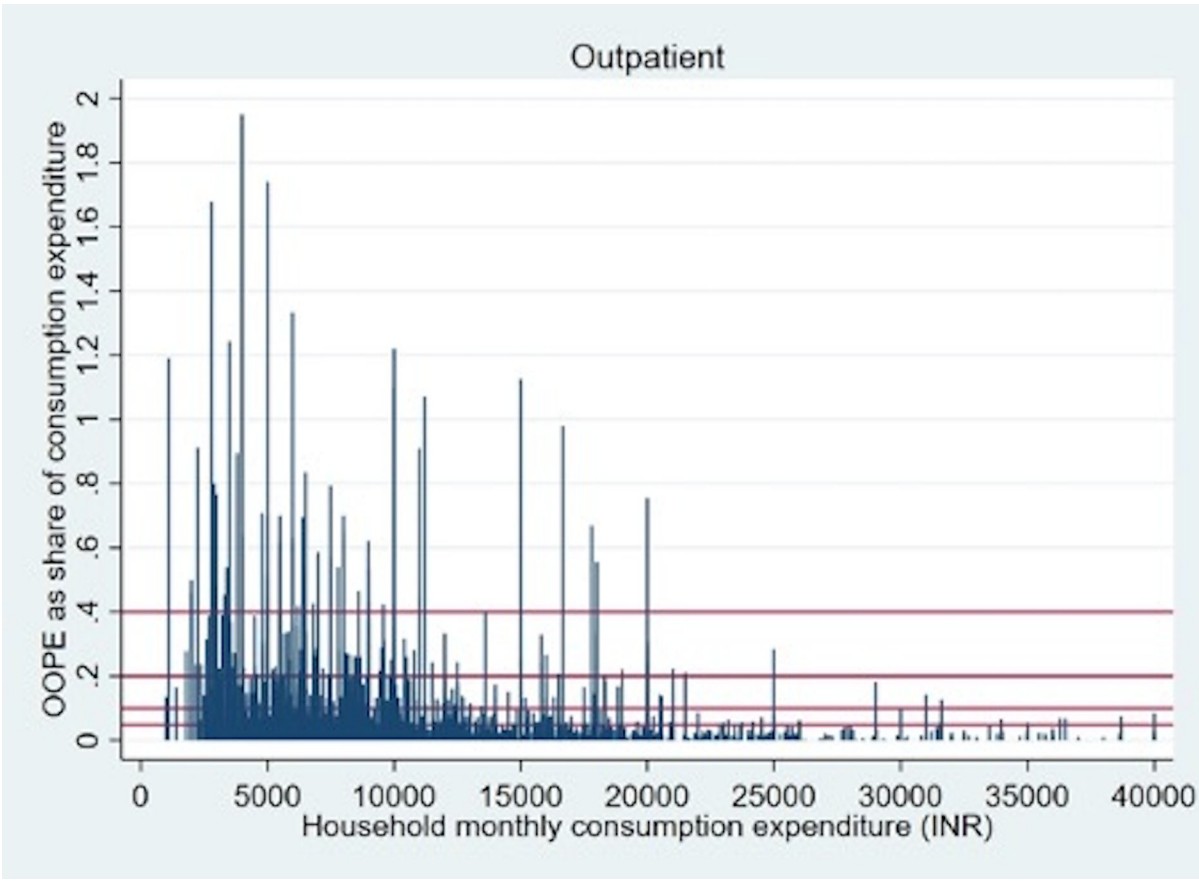

**Fig 1. Outpatient OOPE on childhood infections as a share of per capita monthly consumption expenditure of households.**

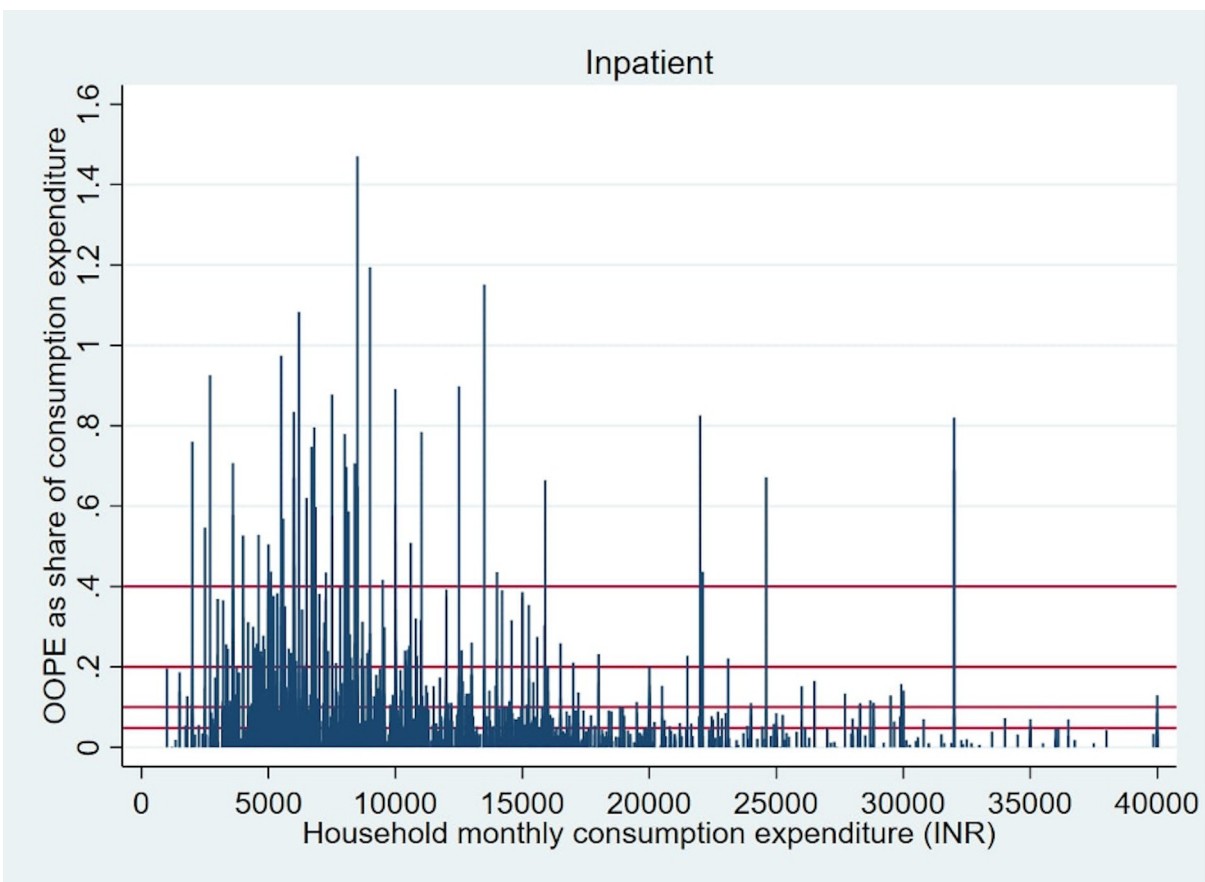

**Fig 2. Hospitalisation OOPE on childhood infections as a share of per capita monthly consumption expenditure of households.**

the households seeking outpatient care for treatment, percentages of households having OOPE higher than 10%, 20% and 40% thresholds are 15%, 5.5% and 1.7% respectively. For hospitalization at the same thresholds percentages of households facing catastrophic expenditure are 20%, 6.4% and 2% respectively. Furthermore, poorer households reported a higher OOPE share of total consumption expenditure as reflected by the concentration of spikes on the left part of Figs

**Table 5. Average share of household consumption expenditure incurred on treatment of childhood infections (age 0–5 years) in outpatient and hospitalisation by different quintiles.**

| Quintile groups of households* | Outpatient | | Hospitalisation | |
| --- | --- | --- | --- | --- |
| | OOPE share (%) on outpatient with 15-days recall | Difference in OOPE share with Richest quintile | OOPE share (%) on hospitalization with 365-days recall | Difference in OOPE share with Richest quintile |
| Poorest 20% (Q1) | 7.9 | 5.1 (<0.001) | 8.2 | 3.2 (0.001) |
| 2nd poorest 20% (Q2) | 5.4 | 2.6 (<0.001) | 10.5 | 5.5 (<0.001) |
| Middle 20% (Q3) | 4.5 | 1.7 (<0.001) | 5.6 | 0.6 (0.503) |
| 2nd richest 20% (Q4) | 3.9 | 1.1 (0.001) | 6.1 | 1.1 (0.211) |
| Richest 20% (Q5) | 2.8 | Reference | 5 | Reference |
| Overall | 4.8 | | 6.7 | |

Note

* based on households' total consumption expenditure

Source: Authors estimates using SCH 2017–18

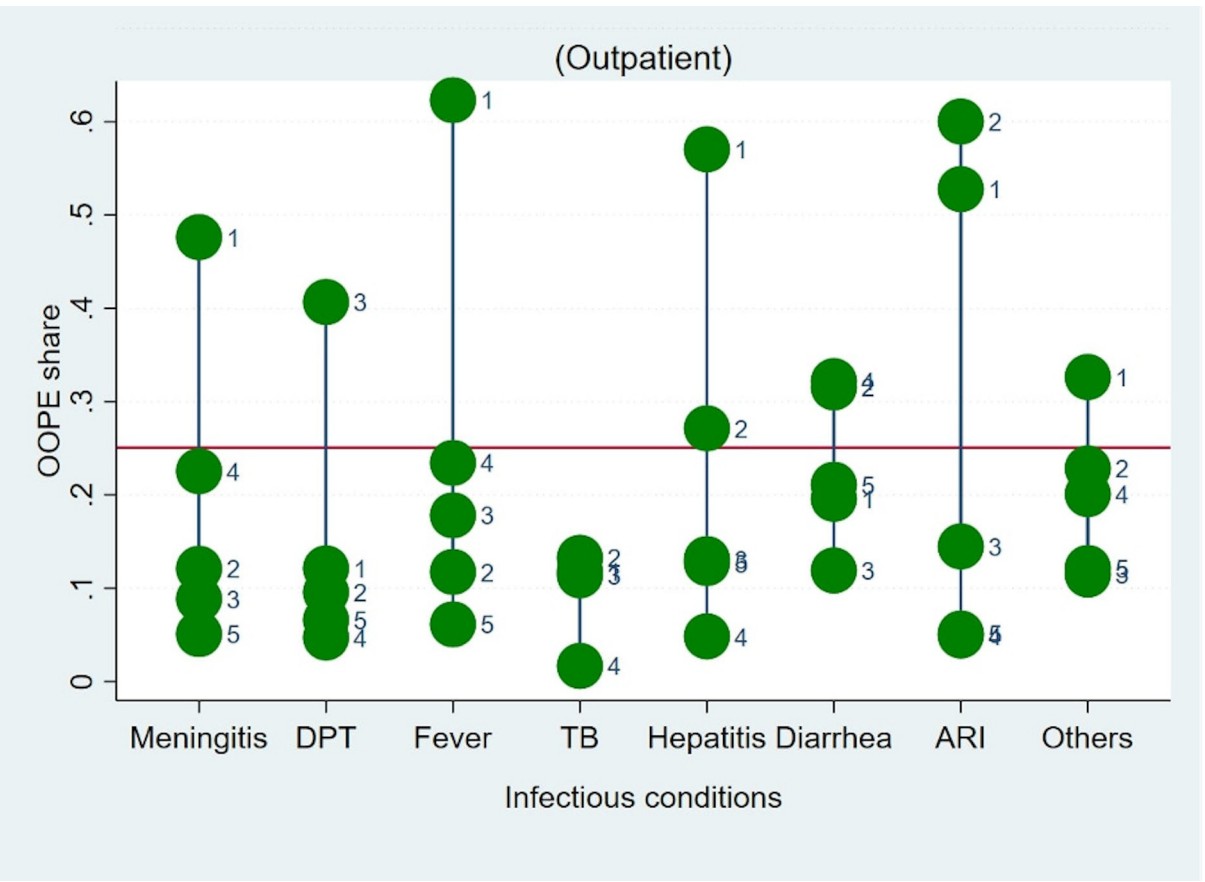

**Fig 3. Outpatient OOPE as a share of household total consumption expenditure on different childhood infections by quintile.** Notes: Horizontal red line represents mean OOPE as a share of household total consumption expenditure; Q1–1st quintile, Q2–2nd quintile, Q3–3rd quintile, Q4–4th quintile, Q5–5th quintile.

1 and 2. Detailed information on OOPE share on specific diseases crossing 20% and 40% of households' total consumption expenditure is presented in S1 Figs A-I and A-II in S1 File.

Lastly, we estimated the extent of the OOPE burden across consumption expenditure quintile groups of households. Table 5, demonstrates that the disease-related OOPE burden was disproportionately higher for the poorest 20% of households (outpatient, 7.9%; hospitalization, 8.2%) in comparison to the 20% richest quintiles (outpatient, 4.8%; hospitalization, 6.7%). Figs 3 and 4, graphically depicts OOPE across selected disease conditions and across MPCE quintiles. It clearly demonstrates a disproportionately high burden of OOPE in poorer households in comparison to richer households for all disease conditions. We also estimated OOPE as a share of household consumption expenditure across households having access to any health insurance. The "unadjusted" estimates suggest households with access to any type of financial coverage, particularly poor households, have marginally lower OOPE as a share of household consumption expenditure on treatment as compared with those who do not. (see S10 Table in S1 File for further details).

## Discussion

We used nationally representative household survey data from NSSO and a rigorous methodological approach to generate OOPE estimates on outpatient care and hospitalization as well as

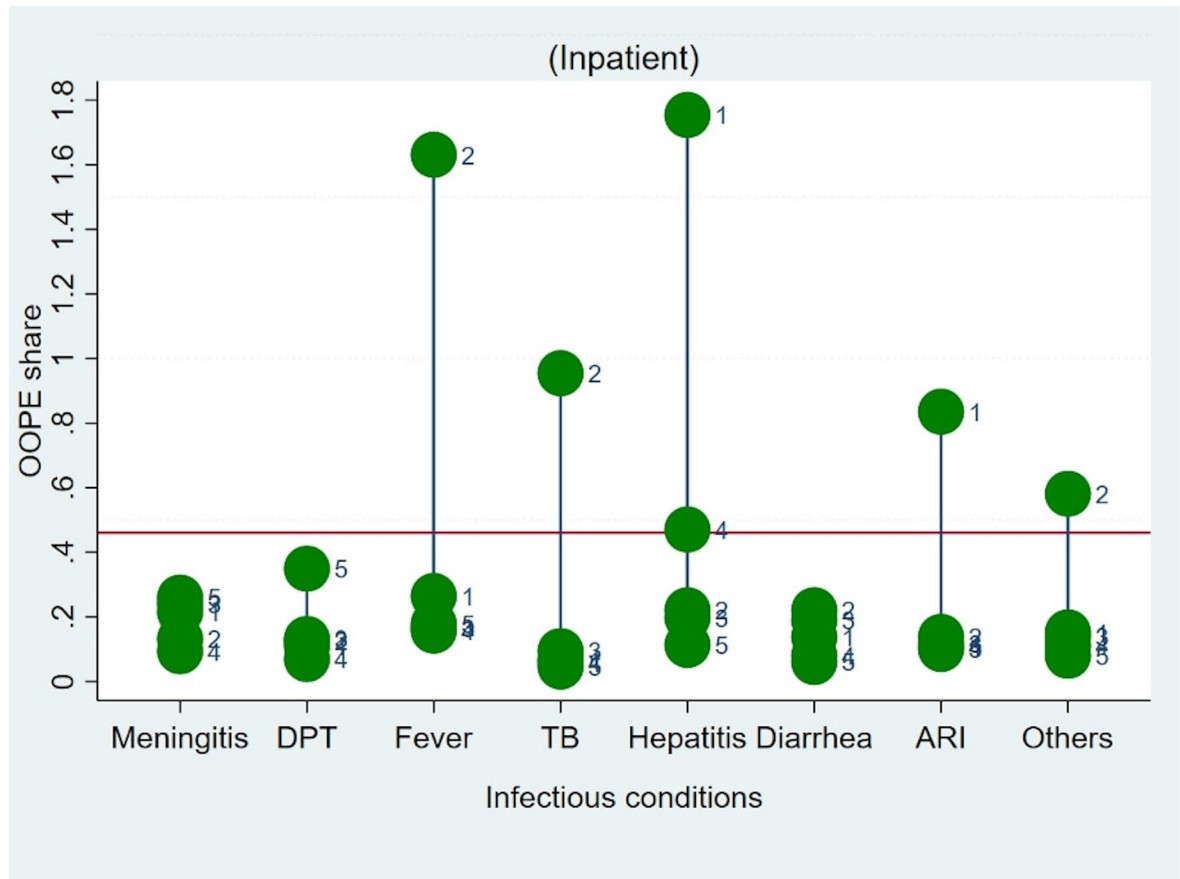

**Fig 4. Hospitalisation OOPE as a share of household total consumption expenditure on different childhood infections by quintile.**
Notes: Horizontal red line represents mean OOPE as a share of household total consumption expenditure; Q1: 1st quintile, Q2: 2nd quintile, Q3: 3rd quintile, Q4: 4th quintile, Q5: 5th quintile.

disease conditions. In addition, we also estimated OOPE as a share of household consumption expenditure by MPCE quintiles to report the quantum of the financial burden borne by Indian households on account of childhood infections. The core strength of our research is the national representativeness of the OOPE estimates and their disaggregation at the level of outpatient care and hospitalization, public and private facilities, disease conditions and income quintiles.

Our estimates suggest that childhood infections continue to be an important cause of health and economic distress in India as reflected by the high prevalence of infections and associated OOPE. The outpatient consultation rate was the highest for fever with rash and acute upper respiratory infections whereas hospitalization rates were highest for fever with rash and diarrhoea. Our findings are in concurrence with published estimates of the burden of childhood infections in India. As per India's State-Level Disease Burden Initiative, the two leading causes of disease burden were diarrhoeal diseases, and lower respiratory infections [17]. In addition, our analysis suggests that acute diarrhoea and ARI remain major causes of hospitalization, and hospitalization-related direct medical expenditure was higher in the private sector health facilities in comparison to the public sector. Previous research has indicated that those patients who consult private healthcare facilities incur higher treatment costs because of diagnostic investigations [18] and medicines. Earlier research has also indicated that medicines are a major contributor to the total treatment cost in the private sector and has a catastrophic impact on

households [9]. Our estimate on component-wise (doctors' fee, medicine, diagnostics, etc) breakup confirms that medicines remain one of the major drivers of the treatment cost in private facilities.

Our estimates also suggest that OOPE as a share of total household consumption expenditure was high (12.5%) even though TB treatment is free at the point of care through the Revised National Tuberculosis Control Program [4]. Though published estimates on TB treatment costs confirm our finding [19, 20], one possible explanation for high OOPE on TB could be the delay in seeking care resulting in hospitalization and consequent OOPE. Other childhood infections that lead to high hospitalization rates and consequently high OOPE are fever with rash and viral hepatitis. The most common causes of viral hepatitis in India are Hepatitis A and Hepatitis E, which primarily spreads through contaminated water [21]. High burden of viral hepatitis-related hospitalizations and associated OOPE on treatment can be attributed to limited access to safe drinking water and poor sanitation in certain sections of the population, especially in remote villages and urban slums.

Furthermore, we observed that OOPE on the treatment of childhood infections constitutes up to 4% to 7% of a household's total consumption expenditure and poorer households face a greater burden of such expenditure, sometimes as high as 20% to 40% of their total consumption expenditure. This treatment-related OOPE is also disproportionately higher for poorer households in comparison to rich, across all disease conditions. Our estimates also suggest that approximately 15–20% of children reporting infectious diseases were not treated in the formal healthcare system (S4 Table in S1 File). Lack of health awareness and lack of financial resources may be the reasons for poor health-seeking by low-income households [22]. Global Enteric Multicenter Study (GEMS) also reported lower expenditure on treatment of diarrhoea by low-income households and low treatment expenditure per episode for girls in comparison to boys from other South Asian [15] and African countries [23].

Though several studies from low and middle-income countries had reported treatment costs for ARI [13], pneumonia [24–26], TB [27], diarrhoea [23], meningitis [28], malaria [29, 30], and dengue fever [31], majority of these studies were based on the narrow cost of illness approach [25, 32] or had small sample size or were focused on the hospital setting [24] or single disease condition [29, 30]. In addition, while some studies reported treatment costs as a proportion of household income [24] and others reported indirect costs such as productivity losses [13, 25, 29, 30, 32], we observed that none of them reported equity impact of treatment-related OOPE as a share of total consumption expenditure. Ours is the first-ever study from India to report treatment-related OOPE for childhood infections as a share of total household consumption expenditure and their distribution across MPCE quintiles highlighting the equity impact of OOPE on the households.

Our study has a few limitations. First, our analysis and estimates of outpatient care and hospitalization are based on self-reported disease conditions as captured in NSSO survey data using standardized working definitions for each disease condition which may not be a true reflection of disease incidence in the population. However, this doesn't affect our estimates of per episode OOPE and the financial burden on the households. Second, the NSSO survey data is cross-sectional in nature hence we were not able to examine the long-term financial consequences of the OOPE incurred by households because of disease episodes and measures taken by the households for consumption smoothening over time. Third, we cannot comment on the causality of expenditure whether less severe disease or low income is leading to low expenditure in certain households. Finally, the scope of our study is limited to reporting treatment-related OOPE across disease conditions and MPCE quintiles and as a proportion of total consumption expenditure hence we have not estimated productivity losses because of disease episodes.

We argue that OOPE related childhood infections cause severe financial stress on the household budget as reflected in the high proportion of consumption expenditure being spent on treatment and care which can compromise expenditure on food, education and other household priorities. Previous studies [33] have reported distress financing such as the sale of assets and borrowing by households in event of hospitalization. One of the reasons for the sale of assets and borrowing could be low savings in poor households. However, we argue that the unpredictability of infectious disease-related hospitalization and consequent OOPE when compounded with a household's inability to smoothen consumption expenditure results in an economic shock leading to asset sales or borrowing. Furthermore, in absence of financial risk protection measures, in extreme cases, households defer or do not seek treatment to avoid financial catastrophe [11] which may lead to poor health outcomes and perpetuation of poor health leading to a poverty cycle.

In summary, our results suggest that though per episode OOPE on outpatient care and hospitalization varied widely across childhood infections and by choice of service provider, these OOPE were a major drain on household resources. We recommend strengthening the public health system and services to ensure free diagnostics and medicines for vulnerable populations. Furthermore, the scale and scope of the government-sponsored health insurance schemes need to be expanded to ensure that private sector services are available to poor households without any financial implications. Ensuring access to care with financial risk protection is urgently needed to address the economic burden of childhood infections on households.

## Supporting information

**S1 File. Supplementary working definitions and analysis.**
(DOCX)

## Author Contributions

**Conceptualization:** Habib Hasan Farooqui, Anup Karan, Sakthivel Selvaraj.

**Data curation:** Habib Hasan Farooqui, Anup Karan, Suhaib Hussain.

**Formal analysis:** Suhaib Hussain.

**Investigation:** Anup Karan.

**Methodology:** Habib Hasan Farooqui, Anup Karan.

**Resources:** Manu Raj Mathur, Sakthivel Selvaraj.

**Supervision:** Anup Karan, Manu Raj Mathur, Sakthivel Selvaraj.

**Validation:** Anup Karan, Sakthivel Selvaraj.

**Writing – original draft:** Habib Hasan Farooqui, Anup Karan, Manu Raj Mathur, Suhaib Hussain.

**Writing – review & editing:** Habib Hasan Farooqui, Anup Karan, Manu Raj Mathur, Sakthivel Selvaraj.

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
