## [Decision Letter · Decision Letter 0]

20 Sep 2022

PONE-D-22-18773Out-of-pocket expenditure on childhood infections and its financial burden on Indian households: Evidence from nationally representative household survey (2017-18)PLOS ONE

Dear Dr. Mathur,

Thank you for submitting your manuscript to PLOS ONE. After careful consideration, we feel that it has merit but does not fully meet PLOS ONE’s publication criteria as it currently stands. Therefore, we invite you to submit a revised version of the manuscript that addresses the points raised during the review process.

We look forward to receiving your revised manuscript.

Kind regards,

Samir Garg, Ph.D

Academic Editor

PLOS ONE

Journal Requirements:

Reviewers' comments:

Reviewer's Responses to Questions

**Comments to the Author**

1. Is the manuscript technically sound, and do the data support the conclusions?

Reviewer #1: Yes

Reviewer #2: Yes

2. Has the statistical analysis been performed appropriately and rigorously? 

Reviewer #1: Yes

Reviewer #2: No

3. Have the authors made all data underlying the findings in their manuscript fully available?

Reviewer #1: Yes

Reviewer #2: Yes

4. Is the manuscript presented in an intelligible fashion and written in standard English?

Reviewer #1: Yes

Reviewer #2: Yes

5. Review Comments to the Author

Reviewer #1: This is an interesting paper. I enjoyed reading the paper that identified newer analyses and domain.

I have some minor observations

1. Estimates may be supplemented with 95% CI wherever applicable

2. Household consumption expenditure is in 30 days reference period and OOP is of 15 days reference. Similarly, hospitalisation is 365 days reference period. May need to mention in limitations

3. The paper used household consumption rather than per capita consumption expenditure. can give one table based on per capita as well as that will be standardised

4. It used wealth index in description but I think authors used MPCE quintile. Pl correct it

5. Focus on implication of the findings in discussion

Reviewer #2: The present study estimates the direct medical and non-medical expenditure incurred by the households on childhood infections and equity related impact. I congratulate the authors in successfully undertaking this analysis. Although, overall, the study methodology and analysis looks great, I have a few concerns as follows:

1. The analysis reports and highlights higher OOPE (both for outpatient and inpatient care) in private facilities as compared to public hospitals. Though, this is an apparent finding, because in the context of India, whole of the treatment expenses are paid out of pocket for getting treatment in private hospitals (if a patient does not have any sort of pre-payment risk pooling mechanism), whereas in public hospitals, treatment is subsidised by the government and thus only some of the proportion of the total expenditure is paid out of pocket. It would be nice to provide component wise breakdown (Doctor fee, medicine, diagnostics, travel, food, etc.) of the direct medical and non-medical expenditure, separately for public and private hospitals. This would point out that on which specific component, the proportion of OOPE is the highest. This would also support one of the conclusions of the study that “public health system needs strengthening in terms of diagnostics and medicines”.

2. The authors did not estimate the impact of insurance on treatment expenses incurred by the families. Considering, the presence of various state level and centrally sponsored public health scheme (even before the launch of AB-PMJAY) across India during the time of data collection, it will be exciting to see whether those insured households had a lower OOP expenses or better financial risk protection as compared to uninsured ones during the event of hospitalization. Further, it will also be interesting to have some discussion around this issue.

On one side, where public hospitals are being funded through supply side financing mechanisms and also strengthened through demand side mechanisms (publicly sponsored insurance), the households are still incurring OOP expenses. Even in private hospitals, those covered through insurance (especially the poorest quintiles), are also not expected to pay any money from the pocket. Considering that the Government is currently more focussed and investing a huge sum of money in the publicly sponsored insurance schemes, it would be great if authors can comment a bit around the impact of insurance in terms of reduction of OOPE. The authors report that poorer have higher extent of financial risk. It would also be interesting to see the effect of insurance specifically among the poorer sections.

3. The authors had used ‘OOPE as a share of household’s total consumption expenditure’ as a measure of financial risk protection. While calculating financial risk, the standard approach is to express OOPE as a proportion of a household’s capacity-to-pay, which is typically represented by non-food consumption expenditure. Since, the richer households often tend to spend a higher proportion of their total expenditure on health, the measure used in this study can be pro-rich. Though, I am fine with the approach used by authors, I would still recommend using the standard methodology.

4. In table 5, the authors show that the disease related OOPE burden was disproportionately higher for the poorest 20% of households (outpatient, 7.9%; hospitalization, 8.2%) as compared to richer quintiles. I would recommend providing a ‘p value’ and showing whether this difference in OOPE across the quintile groups was statistically significant or not. Similarly, a ‘p-value’ for showing difference in OOP across quintile in each of the disease group in fig 2a and 2b should be provided. Also recommend providing p value for table 4.

5. It would be nice to add SE (standard error) alongside mean OOPE in table 2 and table 3.

6. The figures 1a, 1b and 2b, shows that share of OOP expenses among some of the households were even more than the 100% of their consumption expenditure. The authors could also report on the various coping mechanisms undertaken by these households for dealing with such expenses incurred.

7. To make analysis more comprehensive, it would be nice add a section showing urban-rural differences in term of prevalence, utilization and OOPE due to the childhood infectious diseases.

8. In the section of ‘Statistical analysis’, the authors mention that: “We estimated total OOPE for households by summing up OOPE on all episodes of outpatient visits and hospitalisation, for children aged less than 5 years, and averaged over 30 days”. Was the averaging done for those specific cases that had both outpatient and hospitalization event? Not very clear.

9. In the results section in fig 1a and 1b, OOPE as a proportion of consumption expenditure across different levels of thresholds of more than 10%, 20% or 40% of total household consumption expenditure has been reported. It would be more informative to provide an exact estimate of proportion of households crossing the different levels of threshold (i.e., 10%, 20% and 40%). Lastly, it would also make sense, to run a regression analysis (may be logistic regression) to examine the risk of incurring OOPE as a proportion of (e.g.,) more than the 40% of household’s capacity to pay, with covariate including the type of illness, presence of insurance, type of provider, income quintiles, sex, age, etc.

6. PLOS authors have the option to publish the peer review history of their article (what does this mean?). If published, this will include your full peer review and any attached files.

Reviewer #1: **Yes: **Sanjay Mohanty

Reviewer #2: No

---

## [Author Response · Author response to Decision Letter 0]

19 Oct 2022

5. Review Comments to the Author

Reviewer #1: 

This is an interesting paper. I enjoyed reading the paper that identified newer analyses and domain.

Response: Thank you for liking our paper and constructive comments.

I have some minor observations

1. Estimates may be supplemented with 95% CI wherever applicable

R: Thanks for this suggestion. The revised version now incorporates 95% CI for relevant estimates (Table 1 page number 5 and table 4 page number 9).

2. Household consumption expenditure is in 30 days reference period and OOP is of 15 days reference. Similarly, hospitalisation is 365 days reference period. May need to mention in limitations

R: Thank you for this valid and important observation. This is not a limitation to our study as we have estimated OOPE burden separately for outpatient care and hospitalization. In our analysis wherever we used household consumption expenditure as a denominator to OOPE, we standardized estimates of OOPE (across inpatient 365 days and outpatient 15 days) to 30 days. Moreover, the statistical analysis section in the discussion has been appropriately updated to reflect different reference periods for consumption expenditure, outpatient and hospitalization (page number 5). 

3. The paper used household consumption rather than per capita consumption expenditure. can give one table based on per capita as well as that will be standardised

R: We have not used mean household consumption expenditure in any of the tables. We either calculated per episode OOPE (tables 2 and 3) or estimated OOPE as a share of household consumption expenditure (tables 4 and 5). In tables 4 and 5, the per capita OOPE is not relevant as it gets cancelled out because it is part of both the numerator and denominator. However, we have rescaled the x-axis of figures 1A and 1B on the basis of household per capita consumption expenditure.

4. It used wealth index in description but I think authors used MPCE quintile. Pl correct it

R: Thank you for pointing out this inadvertent error. The error has been now corrected (page numbers 9-12)

5. Focus on implication of the findings in discussion

R: Thank you for the feedback. The discussion has been updated and now reflects on the implications of the findings. The update paragraph is as follows 

“We argue that OOPE related childhood infections cause severe financial stress on the household budget as reflected in the high proportion of consumption expenditure being spent on treatment and care which can compromise expenditure on food, education and other household priorities. Previous studies [33] have reported distress financing such as the sale of assets and borrowing by households in event of hospitalization. One of the reasons for the sale of assets and borrowing could be low savings in poor households. However, we argue that the unpredictability of infectious disease-related hospitalization and consequent OOPE when compounded with a household’s inability to smoothen consumption expenditure results in an economic shock leading to asset sales or borrowing. Furthermore, in absence of financial risk protection measures, in extreme cases, households defer or do not seek treatment to avoid financial catastrophe [11] which may lead to poor health outcomes and perpetuation of poor health leading to a poverty cycle. (page numbers 13)

Reviewer #2: 

The present study estimates the direct medical and non-medical expenditure incurred by the households on childhood infections and equity related impact. I congratulate the authors in successfully undertaking this analysis. Although, overall, the study methodology and analysis looks great, 

Response: Thank you for your positive feedback and critical and constructive comments.

I have a few concerns as follows:

1. The analysis reports and highlights higher OOPE (both for outpatient and inpatient care) in private facilities as compared to public hospitals. Though, this is an apparent finding, because in the context of India, whole of the treatment expenses are paid out of pocket for getting treatment in private hospitals (if a patient does not have any sort of pre-payment risk pooling mechanism), whereas in public hospitals, treatment is subsidised by the government and thus only some of the proportion of the total expenditure is paid out of pocket. It would be nice to provide component wise breakdown (Doctor fee, medicine, diagnostics, travel, food, etc.) of the direct medical and non-medical expenditure, separately for public and private hospitals. This would point out that on which specific component, the proportion of OOPE is the highest. This would also support one of the conclusions of the study that “public health system needs strengthening in terms of diagnostics and medicines”.

R: We agree that there is higher OOPE in the private sector compared to that in public is an apparent finding as treatment is subsidised by the government and past literature has already reported this. As suggested, we have now incorporated detailed tables reflecting a component-wise breakdown of expenditures across public and private hospitals in the supplementary material as Table S8 and S9 (page number 5 in the supplementary file) and also mentioned the new findings briefly in the results. We have also updated the results section in light of the additional analysis (page number 7)as follows

 “Private sector expenditure in outpatient care was driven by medicines whereas in hospitalization it was driven by medicines and bed charges. However, in the public sector, the expenditure was driven by medicines for both outpatient care and hospitalization and non-medial expenditure such as food and lodging for patients and attendants in case of hospitalization. Further details about component-wise breakdown (such as doctor fee, medicine, diagnostics, travel, food, etc.) of the direct medical and non-medical expenditure, separately for public and private hospitals is provided in S8 and S9 Table.”

2. The authors did not estimate the impact of insurance on treatment expenses incurred by the families. Considering, the presence of various state level and centrally sponsored public health scheme (even before the launch of AB-PMJAY) across India during the time of data collection, it will be exciting to see whether those insured households had a lower OOP expenses or better financial risk protection as compared to uninsured ones during the event of hospitalization. Further, it will also be interesting to have some discussion around this issue.

On one side, where public hospitals are being funded through supply side financing mechanisms and also strengthened through demand side mechanisms (publicly sponsored insurance), the households are still incurring OOP expenses. Even in private hospitals, those covered through insurance (especially the poorest quintiles), are also not expected to pay any money from the pocket. Considering that the Government is currently more focussed and investing a huge sum of money in the publicly sponsored insurance schemes, it would be great if authors can comment a bit around the impact of insurance in terms of reduction of OOPE. The authors report that poorer have higher extent of financial risk. It would also be interesting to see the effect of insurance specifically among the poorer sections.

R: This is an interesting observation. We agree that access to health insurance may reduce treatment expenditures for households and the presence of several government-funded health insurance schemes in India has a strong potential to reduce the burden of OOPE for pediatric treatment for households. 

However, we feel that estimating the impact of health insurance on OOPE needs a completely different research design which should be able to address a lot of confounders, endogeneity and self-election. For instance, households with higher health risks may be more prompt to enrol in health insurance schemes. high Just comparing the OOPE across cross-section households with and without insurance will not provide enough insights into the issue. We feel that although this is a very interesting topic to work with, this could be altogether a different study and beyond the scope of the present research. Earlier one of the authors of the present study, using rigorous impact evaluation method, published the impact of Rashtriya Swasth Bima Yojana (https://pubmed.ncbi.nlm.nih.gov/28376358/ ) and the benefit incidence analysis of the NHM (https://equityhealthj.biomedcentral.com/articles/10.1186/s12939-021-01489-0 ). 

Nonetheless, we have now included “unadjusted” estimates of OOPE as a share of household consumption expenditure across households having access to any health insurance as supplementary materials as Table S10 (page number 9 in the supplementary file). We find that households with access to any type of financial coverage, particularly poor households, have marginally lower OOPE as a share of household consumption expenditure on pediatrics treatment as compared with those who do not have access to such coverage. We have also updated the results to reflect these important issues. (page number 10)

3. The authors had used ‘OOPE as a share of household’s total consumption expenditure’ as a measure of financial risk protection. While calculating financial risk, the standard approach is to express OOPE as a proportion of a household’s capacity-to-pay, which is typically represented by non-food consumption expenditure. Since, the richer households often tend to spend a higher proportion of their total expenditure on health, the measure used in this study can be pro-rich. Though, I am fine with the approach used by authors, I would still recommend using the standard methodology.

R: We very much appreciate this comment. We agree that ‘OOPE as a proportion of a household’s capacity-to-pay’ is one important standard method of estimating financial risk. Unfortunately, the NSSO 75th Round data does not provide information on the food and non-food breakup of the household consumption expenditure, restricting us to estimate total consumption expenditure and hence the capacity-to-pay of a household cannot be estimated from this data. 

However, literature has widely used other measures such as 10%, 20% ….. etc. of the total household consumption expenditure as well (Xu et al. 2003, 2007 https://www.sciencedirect.com/science/article/abs/pii/S0140673603138615; Karan et al. 2013 https://journals.plos.org/plosone/article/file?id=10.1371/journal.pone.0105162&type=printable; Karan et al 2017 https://www.ncbi.nlm.nih.gov/pmc/articles/PMC5710570/). 

Moreover, for measuring catastrophic OOPE, we have actually used a continuous measure (Figure 1A and Figure 1B). This enables the reader to visualize the variation in the catastrophic OOPE by per capita consumption expenditure of households. 

4. In table 5, the authors show that the disease related OOPE burden was disproportionately higher for the poorest 20% of households (outpatient, 7.9%; hospitalization, 8.2%) as compared to richer quintiles. I would recommend providing a ‘p value’ and showing whether this difference in OOPE across the quintile groups was statistically significant or not. Similarly, a ‘p-value’ for showing difference in OOP across quintile in each of the disease group in fig 2a and 2b should be provided. Also recommend providing p value for table 4.

R: Thank you for the suggestion. A ‘p value’ and showing whether this difference in OOPE across the quintile groups was statistically significant or not is now added to table 5 (page number 10). We have now provided 95% CI for table number 4 (page 9). Furthermore, all other tables have been updated with 95%CI and SE.

5. It would be nice to add SE (standard error) alongside mean OOPE in table 2 and table 3.

R: Thank you for the suggestion. We have now updated table 2 and table 3 in accordance with the suggestion. (page numbers 7 and 8)

6. The figures 1a, 1b and 2b, shows that share of OOP expenses among some of the households were even more than the 100% of their consumption expenditure. The authors could also report on the various coping mechanisms undertaken by these households for dealing with such expenses incurred.

R: Thank you for the suggestion. We have also updated the discussion to reflect the coping strategies adopted by the households (page number 13). 

Reviewer 1 has also suggested us add implications of the findings, which we have addressed as follows which covers the discussion around coping mechanism undertaken by the households as reported in previous studies (including one of the authors) 

“Previous studies [33] have reported distress financing such as the sale of assets and borrowing by households in event of hospitalization. One of the reasons for the sale of assets and borrowing could be low savings in poor households, however, we argue that the unpredictability of infectious disease-related hospitalization and consequent OOPE which when compounded by a household’s inability to smoothen consumption expenditure results in an economic shock leading to asset sales or borrowing.”

7. To make analysis more comprehensive, it would be nice add a section showing urban-rural differences in term of prevalence, utilization and OOPE due to the childhood infectious diseases.

R: Thank you for the suggestion. We have now added the S5 table, S6 table and S7 table in the supplementary material to reflect showing urban-rural differences in terms of prevalence, utilization and OOPE.

8. In the section of ‘Statistical analysis’, the authors mention that: “We estimated total OOPE for households by summing up OOPE on all episodes of outpatient visits and hospitalisation, for children aged less than 5 years, and averaged over 30 days”. Was the averaging done for those specific cases that had both outpatient and hospitalization event? Not very clear.

R: We apologize for any confusion on this. We have further clarified this statement. We have averaged OOPE separately for inpatient and outpatient for 30 days. Since the recall periods of inpatient outpatient expenditures are different (365 days and 15 days respectively), Averaging for 30 days was essential to estimate the share of inpatient and outpatient expenses to total household consumption expenditure which is available only with 30 days reference period. (page number 5)

9. In the results section in fig 1a and 1b, OOPE as a proportion of consumption expenditure across different levels of thresholds of more than 10%, 20% or 40% of total household consumption expenditure has been reported. It would be more informative to provide an exact estimate of proportion of households crossing the different levels of threshold (i.e., 10%, 20% and 40%). Lastly, it would also make sense, to run a regression analysis (may be logistic regression) to examine the risk of incurring OOPE as a proportion of (e.g.,) more than the 40% of household’s capacity to pay, with covariate including the type of illness, presence of insurance, type of provider, income quintiles, sex, age, etc.

R: We have now added estimates on the proportion of households crossing the different levels of thresholds 10%, 20% and 40% for all the infectious conditions taken together. The updated paragraph is as follows

“Figure 1A and Figure 1B clearly demonstrate that childhood infections related to OOPE exert a severe financial burden on households as reflected in OOPE overshoot (defined as more than 10%, 20% or 40% of total household consumption expenditure and represented by red lines in Figure 1A and Figure 1B) in comparison to the mean overall household expenditure (a proxy for the household income). Among all the households seeking outpatient care for treatment, percentages of households having OOPE higher than 10%, 20% and 40% thresholds are 15%, 5.5% and 1.7% respectively. For hospitalization at the same thresholds percentages of households facing catastrophic expenditure are 20%, 6.4% and 2% respectively.”

However, we would humbly like to differ on the reviewer’s observation of running a regression analysis to show the association between socioeconomic status (SES) and capacity to pay because of two main reasons: one, fitting SES sort of association with the capacity to pay for treatment, using observational data such as from NSSO, is often fraught with “selection” and such estimates are not actually reliable. Second, establishing SES association of capacity to pay is not the focus of the study and such efforts would require a different research design and econometric modelling, maybe a scope of future study. Again one of the authors of the present study has published a such study in PLoS ONE earlier for OOPE in general (Karan et al. 2013 https://journals.plos.org/plosone/article/file?id=10.1371/journal.pone.0105162&type=printable)

---

## [Editor Report · Decision Letter 1]

9 Nov 2022

Out-of-pocket expenditure on childhood infections and its financial burden on Indian households: Evidence from nationally representative household survey (2017-18)

PONE-D-22-18773R1

Dear Dr. Mathur,

We’re pleased to inform you that your manuscript has been judged scientifically suitable for publication and will be formally accepted for publication once it meets all outstanding technical requirements.

Kind regards,

Samir Garg, Ph.D

Academic Editor

PLOS ONE
---

## [Editor Report · Acceptance letter]

15 Dec 2022

PONE-D-22-18773R1 

Out-of-pocket expenditure on childhood infections and its financial burden on Indian households: Evidence from nationally representative household survey (2017-18) 

Dear Dr. Mathur:

I'm pleased to inform you that your manuscript has been deemed suitable for publication in PLOS ONE. Congratulations! Your manuscript is now with our production department. 

Kind regards, 

on behalf of

Dr. Samir Garg 

Academic Editor

PLOS ONE